# A Feature-Based Model for the Identification of Electrical Devices in Smart Environments

**DOI:** 10.3390/s19112611

**Published:** 2019-06-08

**Authors:** Andrea Tundis, Ali Faizan, Max Mühlhäuser

**Affiliations:** 1Department of Computer Science, Technische Universität Darmstadt, Hochschulstrasse 10, 64289 Darmstadt, Germany; max@tk.tu-darmstadt.de; 2Software AG, Uhlandstraße 12, 64297 Darmstadt, Germany; ali.faizan@softwareag.com

**Keywords:** electrical devices, classification, energy management, machine learning, smart environment

## Abstract

Smart Homes (SHs) represent the human side of a Smart Grid (SG). Data mining and analysis of energy data of electrical devices in SHs, e.g., for the dynamic load management, is of fundamental importance for the decision-making process of energy management both from the consumer perspective by saving money and also in terms of energy redistribution and reduction of the carbon dioxide emission, by knowing how the energy demand of a building is composed in the SG. Advanced monitoring and control mechanisms are necessary to deal with the identification of appliances. In this paper, a model for their automatic identification is proposed. It is based on a set of 19 features that are extracted by analyzing energy consumption, time usage and location from a set of device profiles. Then, machine learning approaches are employed by experimenting different classifiers based on such model for the identification of appliances and, finally, an analysis on the feature importance is provided.

## 1. Introduction

Through the integration and interconnection of software-centered devices, traditional power system, which are typically centered on mechanical and electrical components, are supported by more complex equipment which enable more advanced functionalities in terms of management and control. The result of such evolution which makes those systems more intelligent is named as Smart Grid (SG) [1,2]. The advantage of having software components in the network enables the introduction of more sophisticated mechanisms for monitoring the SG as well as to support, in a more effective way, the decision process for the dynamic energy distribution according to the current use of energy, resource state and weather conditions [3,4,5,6]. Additionally, from the consumer perspective, specific optimization techniques can be exploited for managing the scheduling of the device usage (for example based on specific hours of the day or week according to the electricity costs) in order to save money. Moreover, as an actor can be producer and consumer (so called prosumer) of electricity in a SG [7], such flexibility enables a different way to distribute the energy and to deal with unexpected emergency situations, resulting from faults and failures in the network [8,9,10]. A general overview of a Smart Grid is depicted in Figure 1, which includes City and Buildings, Power Plants, Wind Turbines, Electric Vehicles, Solar Panels and Smart Homes.

In this scenario, an important role is played by Smart Homes (SHs). They are equipped with electrical devices that can be controlled and monitored remotely not only to achieve economic benefits by saving electricity, but also by contributing in the reduction of carbon dioxide emission in the environment [11]. SHs represent the human side of the SG [12]. They provide a new perspective towards the usage of the energy in the everyday life and, in particular, in the relationship between energy utilities and consumers. Typically, the traditional homes have devices that work locally and manually, usually by switching them on/off by pushing a button, with a limited control in terms of their automatic management. A SH, instead, represents the convergence of energy efficient, controllable electrical appliances and real-time access to energy usage data. This combination of device management and smart grid enables proactively managing energy use in ways that are convenient, cost effective, and good for the environment.

To enable such flexibility, the envisioned communication mechanism foresees two main components: (a) a *Smart Grid Manager (SG Manager)*, which has a global view of the network. It is responsible for making decisions about the energy distribution on the basis of the overall available resources; (b) a *Smart Home Controller (SH Controller)*, which represents an interface between the SG and a house. Every *SH Controller* aims, from one side, to retrieve information regarding the electricity consumption in a house and provides it to the *SG Manager*, and from the other side to manage the electrical devices in the house on the basis of habits and rules specified by the user.

Behind the advantages of a more intelligent energy grid management, one of the main challenges for enabling such a pro-active control relies on the automatic recognition, identification and classification of the electrical appliances. This in turn requires facing several factors [13], such as: (i) *power consumption extraction* that is the process of measuring the energy from different devices in order to identify recurrent consumption patterns; (ii) *multi-mode functionality*, this means that some devices can have multiple operation mode which can be misleading for their identification due to such a complex behavior; (iii) *parallel usage*, this is an important factor that has to be faced, since typically more than one device is in operation at the same time; (iv) *similar characteristics* because many devices can present similarities in the way they use the energy (e.g., consumption, charging time); (v) *external effects* because the data could be spoiled by external and random factors, which are not predictable, such as temperature, communication failures, human influences, etc.

In this context, this work proposes a model for the automatic identification of electrical devices, after they are plugged into an electrical socket. It is the basis for the automatic control of whole functionality of an SH which includes applications regarding, for example, the monitoring and control of appliances, the dynamic load management system based on available resources, power saving by scheduling the devices as well as emergency systems in case of faults or failures of specific resources in the network. The model is based on a set of 19 features which are able to characterize different electrical devices and distinguish them from others. They are derived by analyzing three main aspects: (i) *power consumption*: related to the electricity being consumed by a device for a certain period of time, (ii) *working schedule*: which includes the hours of the days and the time duration when the device is turned on/off, and (iii) *location*: which represents the place where an electrical device is connected on the basis of the electrical socket within the house. Then, machine learning techniques are used to experiment the model through different classifiers, by using a dataset of 33 types of appliances [14]. The goodness of the proposed model is evaluated in terms of its accuracy, for the identification and classification of electrical devices, with respect to the existing approaches discussed in Section 2.

The rest of the paper is structured as follows. In Section 2, the related works about automatic devices identification are presented, whereas a background on machine learning techniques is reported in Section 3. Then, the adopted research approach is described in Section 4, whereas the proposed model as well as the features are elaborated in Section 5. The experimental results of the classification are highlighted and discussed in Section 6, whereas Section 7 concludes the paper.

## 2. Related Work

This section discusses the most relevant research efforts and related solutions, which have been proposed for supporting the identification of electrical devices.

In particular, in [15], a middle layer to connect sockets and devices, which is centered on Measurement and Actuation Units (MAUs), is presented. The MAUs monitor and analyze the electrical power consumption of any connected device individually by providing fine grain analysis. The main information for the classification is based on temporal behavior of the appliances, power consumption, shape of the power consumption, and level of noise. Different classifiers have been experimented with, but better performances have been reached by the Random Forest, LogitBoost, Bagging and the Random Committee, which achieved 95.5% accuracy.

In [13], different energy measurements, such as active power, reactive power, phase shift, root mean square voltage and current, by collecting data of each device in an isolated way, are instead considered. This approach aims to provide a plug and play tool to create energy awareness on the basis of real-time energy consumption of electrical devices. Additionally, multi-mode functionality, parallel usage of devices and external effects are also tackled. The difficulties to support the identification of devices which have multi-mode operation compared to those with a single operation mode are discussed. It resulted in an extensive training by deriving a classification model with an accuracy between 94–97%.

In [16], an approach centered on plug-based low-end sensors for measuring the electric consumption at low frequency, typically every 10 s, is presented. In particular, a sensor called PLOGG is used to record a vector of electrical parameters related to the appliance being monitored [17]. However, for each appliance class, a stochastic model is built from the observed consumption profiles of several instances of each class that are used to train the models. A k-NN classification algorithm has been employed on the basis of the identified features by reaching a level of accuracy equal to 85%. However, in [18], a plug and play “smart plug” is investigated. It aims to recognize the consumer appliance category, which is specified according to consumption scales and priorities, based on the employment of specific sensors. It allows for measuring and recording instantaneous energy consumption, by estimating specific parameters of consumer appliances such as the total harmonic distortion and the power factor.

In [19], an approach based on Non Intrusive Appliance Load Monitoring (NIALM) at meter level, to detect whether the device is switched on or off, is discussed. When a change occurs in the overall electrical power signal of the house, the change is analyzed and compared to the already-known patterns available in a database. Another centralized approach, for monitoring power signal, exploits the ZigBee device, which is attached to the main electrical unit [20]. It is used to identify in real time the appliance that contributes to each spike of energy. Another research effort, based on a centralized approach, is described in [21], in which the authors used a custom data collector and, in particular, a power interface oscilloscope and a computer as hardware. It allows for detecting electrical noise to classify electrical devices in homes by exploiting the electrical noise as an additional parameter. However, time series measurements, which represent electrical signatures of different electrical devices, are used in [22] for their identification.

A summary of the above-mentioned related works is reported in Table 1. Two main approaches, to face with automatic identification of electrical devices, emerged from the above related works. One is based on the employment of additional monitoring devices either distributed [15,22] or centralized [19,20,21] which results expensive in terms of money for their installation and hardly scalable; the second one that does not exploit any additional devices, is centered on energy measurements [13], but it lacks in the categorization and formalization of the adopted features. Some of those works used aggregated traces (AT) of multiple devices and attempt to disaggregate energy usage, whereas other works, as in our case, used directly disaggregated traces (DT). For the sake of the completeness of this paper, Table 1 provides a high-level overview of prior works by highlighting, for example, the used parameters, data collection techniques in terms of type of traces and accuracy they achieved.

Other proposals are available in literature, which are not reported and compared in Table 1 because some of them are based on different approaches and/or input data whereas others have different purposes. For example, there have been other works in the context of appliance identification that are centered on both different approaches and input data, such as those based on high frequency conducted electromagnetic interference (EMI) which use Non-Intrusive Load Monitoring (NILM), as described in [23]. It aimed to present some of the key challenges towards exploiting EMI and the dataset of the collected data, which was used in the experiment, is also available online and freely downloadable [24]. However, the work in [25] proposes a technique that aims at identifying anomalous appliances in buildings by using aggregate smart meter data and contextual information in near real time.

In this wide context, our paper is strictly related to those works, reported in Table 1, which dealt with disaggregated traces. In particular, our work stands out from the previous ones because (i) a set of features that characterize electrical devices are proposed and formalizedl (ii) a model, based on their combination, is used to identify and recognize devices when they are plugged into the circuit without additional monitoring devices and based on disaggregated traces, and (iii) high performances in terms of accuracy are reached.

## 3. Background on Machine Learning

Machine learning (ML) is a data analysis technique, based on computational algorithms able to learn directly from the data without a predefined model [26]. In particular, ML techniques aim to identify patterns from which the extracted information is used to make better forecasts, prediction and decisions.

Thanks to the huge amount of available data, ML represents a popular approach that is exploited in several fields, for facing classification-related problems, such as in (i) computational finance for the evaluation of credit risk and algorithmic trading; (ii) image processing and artificial vision for facial recognition, motion detection and object identification; (iii) computational biology for the diagnosis of tumors, pharmaceutical research and DNA sequencing; (iv) energy production for price and load forecasts; (v) automotive, aerospace and manufacturing sectors, for predictive maintenance; and (vi) natural language processing, for speech recognition applications.

Among the variety of existing techniques, an overview of the most popular ones is provided and briefly discussed below, such as Decision Trees (DTs), Support Vector Machine (SVMs), Linear Regressions (LRs), Naive Bayes (NB), Random Forest (RF), Random Committee (RC), Boosting, Bagging, and Artificial Neural Networks (ANNs). Some of them are then selected and taken into account in the evaluation part of the proposal.

### 3.1. Linear Regression and Decision Trees

The goal of linear regression models is to find a linear mapping between observed features and observed real outputs so that, when a new instance is seen, the output can be predicted [27]. Regression is a method of modeling a target value based on independent predictors. This method is mostly used for forecasting and finding out cause and effect relationships between variables. DTs are decision tools based on a tree-model [28,29]. They are navigated from the root to the leaves, each intermediate node represents a decision point and the ramification represents the properties that leads to a particular decision. The predicate that is associated with each internal node, which is used to discriminate among the data, is called “split condition”. When a leaf is reached by navigating the tree, not only is a particular classification associated with the input instance, but, thanks to the path, it is possible to understand the reason for a particular result. A DT should be used when the relations among the various aspects of a specific application context are difficult to explain. In this case, the nonlinear approach of the DT performs better than the Linear Regression.

### 3.2. Support Vector Machines and Naive Bayes

SVMs are linear models for classification and regression problems which are used to solve linear and nonlinear, problems [30,31]. The idea of SVM is based on the definition of a line or a hyperplane which separates the data into classes. Based on given labeled input, the algorithm outputs a hyperplane-based model that is able to classify new instances. Given a set of training examples (training set), each of which is labeled with the class to which the two classes belong, an SVM training algorithm constructs a model that assigns new examples to one of the two classes, thus obtaining a non-probabilistic binary linear classifier. An SVM model is a representation of the examples as points in space, mapped in such a way that the examples belonging to the two different categories are clearly separated by the widest possible space. The new examples are then mapped in the same space and the prediction of the category to which they belong is made on the basis of the side in which it falls. In addition to linear classification, it is possible to use SVM to effectively perform nonlinear classification using the kernel method, implicitly mapping their inputs into a multi-dimensional feature space. NBs belong to a probabilistic family of classifiers [32]. They are centered on the theorem of Bayes, which is based on the assumptions of independence among features. NBs are highly scalable, requiring a number of parameters linear in the number of variables. Furthermore, the method is very efficient for text categorization, which can compete with more advanced methods including SVMs, with appropriate pre-processing.

### 3.3. Random Forest and Random Committee

The Random Forest is a very popular algorithm for feature ranking [33]. It belongs to the Bagging methods (Boostrap Aggregating) and it is based on the use of multiple decisions trees $DT_{k}$, each of which is trained on a subset $S_{k}$ of the training set. Each new instance provided in input is classified by all the *k*-Decision Trees, each of which provides its own classification. A voting mechanism, which can consist of majority vote rule or on the average value gathered from all the k-classifications, is then adopted to establish the final classification based on the most common class in the node. However, the Random Committee builds an ensemble of base classifiers and generates their prediction by averaging of the estimated probability [34]. Each base classifier is based on the same data, but it uses a different random number seed, which makes sense if the base classifier is randomized; otherwise, all classifiers would work in exactly the same way.

### 3.4. Boosting and Bagging

Bagging and Boosting are also ensemble methods [35,36]. The idea of Boosting is to combine “weak” classifiers in order to create a classifier with a better accuracy. In algorithms such as Adaboost, the output of the meta-classifier is given by the weighted sum of the predictions of the individual models. Whenever a model is trained, there will be a phase of repeating the instances. The boosting algorithm tends to give greater weight to the misclassified instances, with the aim of obtaining an improved model on the basis of these latter instances. On the contrary, the Bagging approach aims to reduce variance from models that might have a very high level of accuracy, but typically only with the data, on which they have been trained, which is called over-fitting. It tries to reduce this phenomenon by creating its own variance among the data by sampling and replacing data by testing diverse models called hypothesis.

### 3.5. Artificial Neural Networks

ANNs are particular computational models, which are able to represent knowledge based on massive parallel processing and pattern recognition based on past experience or examples [37]. An ANN is defined through an initial layer on the basis of the available inputs, a final layer which represents the output of the computation and a hidden layer which is defined in terms of potential multi-layers through which the inputs undergo various transformation and calculation steps as long as the final layer is reached and the output is generated. They are computation models inspired by biological networks in which: (i) the information processing occurs at several simple elements that are called neurons; (ii) signals are passed between neurons over connection links; (iii) each connection link has an associated weight, which, in a typical neural net, multiplies the signal transmitted; (iv) each neuron applies an activation function (usually nonlinear) to its net input (sum of weighted input signals) to determine its output signal. By such replicated learning process and associative memory, an ANN model can classify information as pre-specified patterns. A typical ANN consists of a number of simple processing elements called neurons, nodes or units. Each neuron is connected to other neurons by means of directed communication links. Each connection has an associated weight, which represents the parameters of the model being used by the net to solve a problem. ANNs are usually modeled into one input layer, one or several hidden layers, and one output layer.

## 4. Research Approach Description

This section aims to clarify the approach adopted in this research task, which is depicted as a process in Figure 2. The process is designed in three main parts, which are organized in lanes: “Data Management”, “Phase” and “Work-product”. More specifically, the “Data Management” lane is related to the dataset elaboration, the “Phase” lane provides the information about what is done and in which order, whereas the “Work-product” lane illustrates which output is generated and how it is eventually used. By describing the process by phase from the top to the bottom, its functioning as well as the sequence order of its actions are highlighted. In particular:*Model definition*: this phase starts by taking in as input an *Initial Dataset* which contains a collection of data related to different types of devices. This phase aims at identifying common features among the different type of devices that will be used to characterize and discriminate them. The output of this phase is represented by a model based on different features called *Feature-based model*.*Feature-driven value extraction*: this phase uses both the *Initial Dataset* and the *Feature-based model*. In particular, the *Feature-based model* is applied to the *Initial Dataset* and, in particular, on the recorded traces in order to extract additional information, called *Derived Dataset*. Such information, which enriches the traces available in the *Initial Dataset*, is then exploited to distinguish the different appliances.*Data splitting*: this phase is centered on the *Derived Dataset* and aims to divide it into two disjoint subsets: *Training Set* and *Test Set*. The *Training Set* is provided in input to a learning algorithm in order to train appropriately the classifier which is built on the basis of the identified features, whereas the *Test Set* is used to validate it.*Learning*: in this phase, one or more learning algorithms are chosen on the basis of both the analysis to be conducted (e.g., supervised, semi-supervised or non-supervised) and the kind of available data (e.g., labeled or not labeled). This phase ends by training such learning algorithm by using the *Training Set* in order to obtain a *Trained classifier* from each of them.*Model validation*: in this phase, the capacity of the trained classifiers obtained in the previous phase in the identification and classification of electrical devices is assessed. To this aim, only the *Test Set*, which consists of traces of devices that have been not used to train the classifiers, are employed so as to get the *Classification Results*.*Feature analysis*: this phase ends the overall process. In particular, starting from the *Classification Results* gathered from the previous phase, the features that play the most important role, in the electrical devices identification, are discussed.

In the next sections, more in-depth details on the conducted research activity are given, by focusing on the work-products. Specifically, Section 5 provides the full description of the *Feature-based model* by describing a particular instance of it called EDIM (Electrical Devices Identification Model) by highlighting which aspects have been considered in its definition and why, whereas the *Trained classifiers*, the *Classification Results* and the *Importance Feature Results Analysis* are discussed in Section 6.

## 5. EDIM: An Electrical Devices Identification Model and Related Features

In this section, the proposed *Electrical Devices Identification Model (EDIM)* for the identification of electrical devices is described. Specific aspects, related to the usage of appliances in terms of time, place and electricity consumption have been considered for its definition. Their combination aims to highlight emergent behaviors that are able to characterize and discriminate among different devices.

In particular, as it is depicted in Figure 3, the *EDIM* model is inspired by three main driving questions: (i) HOW MUCH does a device consume? (ii) WHEN does a device consume? (iii) WHERE is a device used? The rationale behind them relies not only on extracting and using information that is directly derivable or measurable from a device, such as its energy consumption, but also to combining it with further information related to the way of using a particular device—for example, by considering differences of the use of a device in specific daily time slots, weekly or seasonal, relationships with other devices such as their use in sequence or in parallel, as well as by distinguishing whether a device is used in a specific area or if it is used, with a certain frequency, in different places.

By dealing with the above-mentioned questions, three main related *feature classes* have been identified, namely *Energy and Power Consumption*, *Temporal Usage* and *Appliance Location*. For each of them, a set of specific features have been proposed, for a total of 19 basic features. In the next subsections, the description of the defined *feature classes* is provided and, for each of them, the features that have been proposed are presented and formalized through a mathematical notation.

### 5.1. Energy and Power Consumption Class

This feature class focuses basically on the measurement of power and energy at various levels and at specific points of time. The features belonging to this class aim to extract information related to the electricity consumption of a device in order to characterize it, by answering the question “HOW MUCH does a device consume?” In this class, the following consumption-related features are identified: *Daily Power Consumption*, *Max Power*, *Power Deviation*, *Average Power*, *Average Active Power*, *Lower Activity Power (or MinPower)*, *Energy Consumption*, *Average Peak Value*, *Power Dense Location*, and *Standby Devices*. In the next section, the above-mentioned features, which have been proposed in this class, are elaborated.

*Daily Power Consumption*. It is used to compute the amount of power *p* consumed by a device *j* on a day D, by observing it in a time unit i[s]∈D:(1)$$P_{j}^{Tot}=\sum_{i\in D}p_{j}\left(i\right),\mathrm{where}\;\;i\;\mathrm{corresponds}\;\mathrm{to}\;a\;\mathrm{second}\;\left[s\right].$$

*Max Power*. This feature is used to calculate the maximum power value *p* used by a device *j* within a specific day *D*:(2)$$P_{j}^{Max}=max\left\{p_{j}\left(i\right)\right\},\mathrm{where}\;i\left[s\right]\in D.$$

*Power Deviation*. This feature deals with the power deviation which is computed as the sum of the difference between the *Max Power* of a device *j* within a reference day *D* and its power consumption at every time unit i[s]∈D. Only when *j* is in operation $Status_{j}\left(i\right)$
>λ=5[W]=On:(3)$$P_{j}^{Dev}=\sum_{i\in D}\left(P_{j}^{Max}-p_{j}\left(i\right)\right)\Leftrightarrow \;Status_{j}\left(i\right)=On.$$

*Average Power*. Given a device *j*, this feature calculates the average power used from it, related to a day *D* by considering both active and non-active operation time i∈D:(4)$$P_{j}^{Avg}=\frac{\sum p_{j}\left(i\right)}{count(i)},\forall \;i\in \;D.$$

*Average Active Power*. Given a device *j*, this feature calculates the average power used from it, related to a day *D* by considering only the active operation time i∈D, such that $Status_{j}\left(i\right)$
>λ=5[W]=On:(5)$$P_{j}^{AvgAct}=\frac{\sum p_{j}\left(i\right)}{count(i)},\forall \;i\in D\;s.t.\;Status_{j}\left(i\right)=On.$$

*Lower Activity Power (Min Power)*. This feature is used to calculate the minimum power value *p* used by a device *j* within a specific day *D*, by considering only the active operation time i∈D, such that $Status_{j}\left(i\right)$
>λ=5[W]=On:(6)$$P_{j}^{Min}=min\left\{p_{j}\left(i\right)\right\},\mathrm{where}\;i\left[s\right]\in D\;s.t.\;Status_{j}\left(i\right)=On.$$

*Energy Consumption*. Given a time period *D* (e.g., a Day, a Week, a Month) divided into a set of *n* sub-periods $\{d_{1},d_{2},\ldots ,d_{n}\}\subset D$. This feature is used to calculate the energy consumption of a device *j* in a specific sub-period b∈D:(7)$$EC_{j}^{b}=\sum_{i}^{D}p_{j}\left(i\right)\Leftrightarrow \;i\;\in \;b\subset D.$$

*Average Peak Value*. Given a reference period of time $D_{j}$ (e.g., a Day, a Week, a Month) as a disjoint list of K={1,2,…,k} time intervals $I_{j}$ = {$sp_{j}\left(h_{0},h_{1}\right)$, $sp_{j}\left(h_{2},h_{3}\right)$, …, $sp_{j}\left(h_{k-1},h_{k}\right)$} in which a device *j* was actively used, then the Average Peak Value of a device *j*
$APV_{j}$ calculates the average of all the peak values within the considered period of time $D_{j}$, where $peak\left(sp_{j}\left(h^{{}^{\prime }},h^{{}^{\prime \prime }}\right)\right)=max\left\{p_{j}\left(i\right)\right\}$ with $h^{\prime }<i<h^{\prime \prime }$ is the max value of energy consumed from the the device *j* in the time interval $\left[h^{\prime },\;h^{\prime \prime }\right]$:(8)$$APV_{j}\left(D_{j}\right)=\frac{\sum peak(sp_{j}\left(h^{{}^{\prime }},h^{{}^{\prime \prime }}\right))}{k},\;\forall \;sp_{j}\left(h^{{}^{\prime }},h^{{}^{\prime \prime }}\right))\in \;I_{j}.$$

*Power Dense Location*. Given a location *l* and a set of *n* devices J = $\left\{j_{1},j_{2},\ldots ,j_{n}\right\}$. This feature provides the amount of power consumed in *l* from all the devices in J in an arbitrary period of time D, if the total power consumed is more than a reference threshold $PD_{threshold}$:(9)$$PD_{D}^{l}=\sum_{j}^{J}\sum_{i=0}^{T}p_{j}^{l}\left(i\right)>PD_{threshold}.$$

*Standby Devices*. Given a set of *n* devices J = $\left\{j_{1},j_{2},\ldots ,j_{n}\right\}$, this feature calculates a subset of devices $SB_{Dev}=\left\{j_{1},j_{2},\ldots ,j_{k}\right\}\subset J$ which are neither Off nor On, rather those which present a standby mode that is a power consumption $0<p_{j}\left(i\right)<\lambda =5\left[W\right]$ for at least an uninterrupted period of time δt:(10)$$SB_{Dev}=<j_{1},j_{2},\ldots ,j_{k}>,\mathrm{if}\;\exists \;p_{j}\left(i\right)\;s.t.\;0<p_{j}\left(i\right)<\lambda ,\;\mathrm{and}\;continuous\left(i\right)>\delta t.$$

### 5.2. Temporal Usage Class

This feature class focuses on the use of a device, mainly from a temporal point of view, by considering the question “WHEN does a device consume?” The features that fall into this class try to extract information, regardless of the amount of energy consumed, with the aim of identifying temporal usage patterns such as daily, weekly, seasonal related to a single device as well as sequence-parallel relationships between multiple devices (e.g., the dryer after the washing machine, or the decoder along with the television). In this class, the following time-related features are identified: *On-Off Time*, *Active Time*, *Average Active Time*, *Active Duration*, *Most often Usage Time*, *Devices Used in Sequence*, *Devices Used in Parallel*. The above-mentioned features, which have been proposed in this class, are described below in more detail.

*On-Off Time*. This feature is used to know, in which instant of time *i* of a day *D*, a device *j* is turned On/Off. The function $Status_{j}\left(i\right)$ is used to check the working status of *j* at the time *i* [s] ∈D, in order to identify when a change occurs, based on the previous instant of time i−1, where λ = 5[W] is the On-threshold:(11)$$T_{j}^{On\text{-}Off}\left(i\right)=\left\{\begin{array}{cc} Off, & when\;{\mathrm{Status}}_{j}\left(i-1\right)>\lambda \;\;and\;{\mathrm{Status}}_{j}\left(i\right)=0,\\ On, & when\;{\mathrm{Status}}_{j}\left(i-1\right)=0\;\;and\;{\mathrm{Status}}_{j}\left(i\right)>\lambda . \end{array}\right.$$

*Active Time*. This feature counts the number of times that a device *j* is turned on in an arbitrary day D. For example, a dishwasher is typically turned on once or twice a day:(12)$$T_{j}^{Act}\left(D\right)=count\left(T_{j}^{On\text{-}Off}\left(i\right)\right)\Leftrightarrow T_{j}^{On\text{-}Off}\left(i\right)=On,\;\forall \;i\in D.$$

*Active Duration*. Given a device *j*, its active duration in an arbitrary day *D* represents the overall time *i* in which *j* is active that is $Status_{j}\left(i\right)=On$:(13)$$T_{j}^{ActiveDur}\left(D\right)=count_{j}\left(i\right)\;\Leftrightarrow \;Status_{j}\left(i\right)>\lambda =5\left[W\right]=On,\;\mathrm{with}\;i\in D.$$

*Average Active Time*. This feature calculates the active average duration of a device *j* within an arbitrary day *D*:(14)$$T_{j}^{AvgAct}\left(D\right)=\frac{T_{j}^{ActiveDur}\left(D\right)}{T_{j}^{Act}\left(D\right)}.$$

*Most Often Usage Time*. Given a reference period of time $D_{j}$ (e.g., a Day, a Week, a Month) as a disjoint list of K={1,2,…,k} time intervals $I_{j}$={$sp_{j}\left(h_{0},h_{1}\right)$, $sp_{j}\left(h_{2},h_{3}\right)$, …, $sp_{j}\left(h_{k-1},h_{k}\right)$} in which a device *j* was actively used, then the most often usage time of a device *j* indicates the longest interval of time $\Delta h_{j}$=$<h^{{}^{\prime }},h^{{}^{\prime \prime }}{>}_{j}$ such that $sp_{j}\left(h^{{}^{\prime }},h^{{}^{\prime \prime }}\right)\in I_{j}$ and $h^{{}^{\prime }},h^{{}^{\prime \prime }}\in K$, in which the device *j* was used:(15)$$\Delta h_{j}\;=max<h^{{}^{\prime }},h^{{}^{\prime \prime }}{>}_{j}=max\left\{{\left(h^{{}^{\prime \prime }}-h^{{}^{\prime }}\right)}_{j}\right\}=max\left\{sp_{j}\left(h^{{}^{\prime }},h^{{}^{\prime \prime }}\right)\right\}.$$

*Devices Used in Sequence*. Given two instants of time $i_{1}$ and $i_{2}$ with $i_{2}\geq i_{1}$. A device $j_{2}$ works in sequence after $j_{1}$
$seq(\left(j_{2},j_{1}\right))$, when $j_{1}$ stops working at time $i_{1}$, which is $Status_{j1}\left(i_{1}-1\right)=On$ and $Status_{j1}\left(i_{1}\right)=Off$ and the device $j_{2}$ starts working in a subsequent instant of time $i_{2}$, which is $Status_{j2}\left(i_{2}\right)=Off$ and $Status_{j2}\left(i_{2}+1\right)=On$. Thus, this feature ${\bar{SEQ}}_{J}\left(D\right)$ returns all the possible couples of devices $j_{1},j_{2}$
∈J, which work in sequence in a reference period D:(16)$${\bar{SEQ}}_{J}\left(D\right)={\left\{seq\left(j_{1},j_{2}\right)\right\}}_{D}\;\forall \;j_{1},\;j_{2}\in J\;\mathrm{set}\;\mathrm{of}\;\mathrm{devices}.$$

*Devices Used in Parallel*. Given two instants of time $i_{1}$ and $i_{2}$ with $i_{1}>i_{2}$, $i_{1}>\left(i_{2}+1\right)-\Delta t$ and Δt≥1. The device $j_{2}$ works in parallel with $j_{1}$
$par(\left(j_{2},j_{1}\right))$, when $j_{1}$ stops working at time $i_{1}$ that is $Status_{j1}\left(i_{1}-1\right)=On$ and $Status_{j1}\left(i_{1}\right)=Off$ and the device $j_{2}$ starts working in an instant of time $i_{2}$ that is $Status_{j2}\left(i_{2}\right)=Off$ and $Status_{j2}\left(i_{2}+1\right)=On$. This feature ${\bar{PAR}}_{J}\left(D\right)$ returns all the possible couples of devices $j_{1},j_{2}$
∈J, which work in parallel at least for a threshold Δt in a reference period D:(17)$${\bar{PAR}}_{J}\left(D\right)={\left\{par\left(j_{1},j_{2}\right)\right\}}_{D}\;\forall \;j_{1},\;j_{2}\in J\;\mathrm{set}\;\mathrm{of}\;\mathrm{devices}.$$

### 5.3. Appliance Location Class

The place of use of a device is another important indicator, since some electrical devices are often used in the same place (e.g., the hairdryer in the bathroom, the kettle in the kitchen). Some of them are movable and others are not. As a consequence, some devices can be used in more than one location inside a house and they can be active in not more than one location at a time. This feature class is driven by the question “WHERE is a device used?” Considering this aspect, in this class, the following location-related features are identified: *Place of Use*, *Sequence of Usage Location*.

*Sequence of Usage Location*. Given a set of locations L=$\left\{l_{1},\ldots ,l_{z},\ldots ,l_{k}\right\}$ that represent specific places (e.g., a kitchen, a bathroom, a bedroom, a living room and so on) in a given house *h*. The feature computes the list of locations in a house *h*, where a device *j* was chronologically used $p_{j}^{l_{z}}\left(i_{w}\right)>\lambda =5\left[W\right]=On$ in an arbitrary day *D*:(18)$${\bar{SoUL}}_{j}^{h}\left(D\right)=<l_{1}\left(i_{1}\right),\ldots ,l_{z}\left(i_{w}\right),l_{z+1}\left(i_{w+1}\right),\ldots ,l_{k}\left(i_{w}\right){>}_{j}^{h},\mathrm{with}\;i_{1}\ldots i_{w}\in D,$$

$\forall \;<i_{w},i_{w+1}>\mathrm{with}\;i_{w}<i_{w+1}\;$ and $p_{j}^{l_{z}}\left(i_{w}\right)>\lambda \;\mathrm{and}\;p_{j}^{l_{z+1}}\left(i_{w+1}\right)>\lambda$.

*Place of Use* Given a set of locations L=
$\left\{l_{1},l_{2},\ldots ,l_{k}\right\}$ that represent specific places (e.g., a kitchen, a bathroom, a bedroom, a living room) in a given house *h*. This feature allows for knowing in which location z∈L the device *j* was used $p_{j}^{z}\left(i\right)>\lambda =5\left[W\right]=On$. That is, it was On for at least one time unit *i* in an arbitrary day D:(19)$$z_{j}^{h}\left(i\right)⇐p_{j}^{z}\left(i\right)>\lambda ,\;\mathrm{with}\;z\in L\;\mathrm{and}\;i\in D.$$

## 6. Experiments and Results Discussion

In this section, first an overview of the used dataset is given, then the results gathered by experimenting the proposed model are described and, finally, a discussion on the importance of the features is provided.

### 6.1. Dataset Overview

The dataset used to evaluate the proposal consists of a collection of traces related to the daily use of different electrical devices. This dataset, which is made available under the Open Database License (ODbL) [38], is public available and freely downloadable [14]. Each entry of the dataset contains basic data such as the identifier of a device, the time unit with a granularity of a second, which is used to collect the data of each device, the amount of energy consumed in a time unit and so on.

The trace-base data have been grouped into three categories in accordance with the process they have been collected: “Full-day traces”: which contains complete traces that have been recorded for a time period over 24 h; “Incomplete traces”: which contains traces with missing measurements in different instants of time over the day; “Synthetic traces”: which contains trace fragments of devices that have been manually completed with values corresponding to zero consumption readings, when the real values were not available. In our case, the folder containing the “Full-day traces” has been used in the experimentation, which corresponds to 33 types of devices, as listed in Table 3.

### 6.2. Features Evaluation

A machine learning based approach has been used for the evaluation of the proposed identification model. Starting from such row data available in the “Full-day traces” folder, additional information has been extracted by using the features that have been proposed in Section 5. Both the basic and extracted information is used to train and test different classifiers. As the dataset is labeled, the case in consideration falls under a supervised learning problem. As a consequence, only supervised machine learning techniques have been considered and compared. In particular, Random Forest, Bagging, LogitBoost, Decision Tree, Naive Bayes and SVM algorithms have been selected and experimented with for two main reasons: (i) on one hand, to the best of our knowledge, they have shown the best performance in literature, (ii) on the other hand, it is possible to analyze and understand the logic behind their classification process, which is, instead, not always possible with other techniques. For example, neural networks make difficult to understand what happens during the classification, since they act as a black box, to understand which features play the most important role for the device identification. As a consequence, they have not been considered. For the sake of completeness and clarity of this work, Figure 4 summarizes the machine learning algorithms that have been used in the experiments along with the values of the parameters that have been used after having tuned them.

The training and test set have been constructed by using a standard approach based on 5-fold cross validation, so as to reduce both the risk of losing important patterns/trends in data set and the error induced by bias. In more detail, 80% of all 33 categories of available devices’ traces have been exploited to build the training set extracted from the traces, with the remaining 20% of the data to build the test set, which are employed respectively to train and test the above-mentioned classifiers. Moreover, the testing of the model was done for each single device in order to get the prediction of each device separately (i.e., at device level), and then the accuracy of all the devices were averaged to calculate the overall accuracy of each trained model. Moreover, as different traces of different models of certain device categories were available (for example LCD-TV, Router, Washing Machine), the construction of the training set and the test set took into account that the traces belonging to a certain device model were used only in the training set or in the test set. This allowed for showing that the proposed features are able to recognize even new devices with similar behavior, in terms of energy usage, belonging to one of the 33 device categories under consideration.

A first result is shown in Table 2 in terms of accuracy of the different classifiers. A general observation is that all the classifiers present an accuracy higher than 90%, which provides an indication of the goodness of the selected features. Indeed, they show a certain degree of independence from a specific classifier and, as a consequence, they are well suited to describe appliance types and distinguish them from others. Moreover, among them the best performance is reached by the Random Forest algorithm with 96.51% accuracy, which is why the rest of the analysis is specifically based on its use for the subsequent evaluation. The details are reported in Table 3 by showing the result values for *true* and *false positives* as well as the *precision*, which measures the proportion of actual negatives that are correctly identified as such, and the *recall*, which measures the proportion of actual positives that are correctly identified as such, for each kind of device.

As we can see, our implementation reaches at least 80% accuracy and almost all the devices are always classified correctly, which can be seen from the true positive ratio, which is equal to 1.0. However, for other devices, the false positive ratio is, however, very low. Additionally, both the results obtained by calculating the precision and the recall values comply with the observed values related to the true positive rate. Only for one device, and in particular for the Water Kettle, a lower level of classification is shown. This is associated with the limited number of instances available during the training phase of the classifier that made the training phase of the classifier very difficult for this typology of device.

In general, some devices present electrical characteristics that are easier to recognize and which require few instances for training the classifier; others instead require a greater number of instances. This can be traced back to the fact that the behavior of some devices is not only strongly dependent on their mode and state of operation, such as for an alarm clock or a vacuum cleaner. Others, instead, have dependencies on their state and external factors, for example in the case of the Water Kettle, the amount of water to be heated could influence the duration of its heating process. Consequently, not only is it necessary to have a sufficient amount of traces, but they should also be collected considering these additional factors.

However, more than 60% of the devices are correctly identified and classified, as we can see from the true positive rate, precision and recall, without errors. In summary, we globally obtained very high performances in terms of accuracy compared with other related works, and in particular with respect to the related work [15], by only using 19 features instead of 517 features on the same dataset, by requiring both less computational resources and computing time.

### 6.3. Discussion of the Features Importance

In addition, not only is it important to have a model that performs well, but it is also very important to understand why it works good (or bad) and under which conditions. This helps to understand the logic of the model and the reasoning behind a decision. Knowing the importance of a feature in the classification process may motivate the exploitation of more complex one or removing them based on their significance, even by sacrificing some accuracy for the sake of the interpretability. In our case, an analysis on the feature importance has been conducted, in order to know which of them plays the most important role in the electrical device identification process.

In the assessment of the feature importance, a common evaluation criterion is called impurity, which is used to express the level of homogeneity (or heterogeneity) among a group of items [39]. In our case, the classification model is based on the Random Forest, which in turn consists of a set of sub-trees. In this case, the impurity value for each feature is calculated on the basis of each sub-tree. As a consequence, the impurity is assessed over all the nodes for all the trees. Every node in the decision tree splits the dataset into two subsets, so that all the results showing similarities fall in the same subset. As a consequence, from one side, the more important a feature is, the more it decreases the impurity in the tree, but, on the other side a mistake based on it will produce a greater impact on the overall classification. Typically, the features which generate greatest decrease in terms of impurity are located closer to the root-level of a tree, whereas those that produce less decrease of impurity are closer to the leaf-level.

In this case, Table 4 shows how much each feature contributes in the reduction of the level of impurity and, as a consequence, their importance in percentage in the identification of the electrical devices. As it is visible, the features are sorted by level of importance. The most important feature is shown in the first row of the table. Moving down, the less important features are listed. In particular, the *Avarage Peak Power* is the most important one because it provides the highest level of impurity reduction, unlike with the *Energy Consumption*, which is the feature that contributes the least to discriminating between appliances and, consequently, to reduce the level of impurity among them. It is important to note that no value of importance is reported regarding to the *On-Off Time*. This is because it does not contribute directly as a discriminating characteristic, but it is used indirectly from other more complex features in the classification process, as a supporting feature.

A further observation can be made by calculating the average percentage value of importance of a feature ${\bar{I}}_{F}=5.26\%$ as:(20)$${\bar{I}}_{F}=\frac{100}{count(F)}=\frac{100}{19}=5.26\%,\;\;\mathrm{with}\;F\;\mathrm{the}\;\mathrm{set}\;\mathrm{of}\;\mathrm{all}\;\mathrm{features}.$$

On the basis of such reference parameter ${\bar{I}}_{F}$, only the first eight features of Table 4 present a higher value than ${\bar{I}}_{F}$. It means that *Average Peak Power*, *Average Active Power*, *Max Power*, *Average Active Time*, *Lowest Active Power*, *Place of Use*, *Active Duration*, *Devices Used in Parallel* can be considered the most important features in the classification process. Indeed, by summing their percentage values of importance, the result is equal to 82.31%, which reflects the level of accuracy of at least 80%, in the classification of almost all the devices with a true positive ratio equal to 1.0, as described above. However, the diagram depicted in Figure 5 reports for such features both the absolute importance values with respect to all 19 features as well as the relative values when only the most important ones are considered.

## 7. Conclusions

The paper focused on the automatic identification of electrical devices based on features. An *Electrical Device Identification Model* centered on three feature classes related to energy consumption, time usage and location, have been proposed. For each class, specific features have been defined and formalized for a total of 19 distinct features. The information extracted by applying such features has been used to train six different classifiers (i.e., Random Forest, Bagging, LogitBoost, Decision Tree, Naive Bayes and SVM), which have shown a high level of accuracy as a symptom of goodness of the proposed features. Of course, a number of variations of techniques, which are designed for different types of tasks, are also available. From one side, they typically allow for obtaining even better results; this means that, to some extent, they influence the assessment. Since our aim was to evaluate the proposed features by avoiding any potential kind of bias, in this research activity, we considered only standard techniques in order to obtain a more neutral assessment of the features and to compare the results with the prior works reported in Section 2. Further experiments and details related to the Random Forest classifier that provided the highest accuracy equal to 96.51% have been conducted and discussed, as it outperformed with respect to the related work [15] by using only 19 features on the same dataset. In particular, the ratios of true and false positives, as well as the precision and the recall with reference to the specific appliances, have been evaluated. An additional analysis has been done, in order to understand the logic of the classifier and the reasoning behind its decisions. Specifically, not only did it emerged how much each feature contributes in the classification process, but also the most important ones have been identified.

Ongoing works aim to extend this identification model in order to (i) enhance the interaction between Smart Homes and Smart Grids to improve the decision-making process for the energy automatic management and distribution in the network, and (ii) improve the local management of electrical devices in smart homes automatically on the basis of users’ habits, and centered on the definition of specific user’ profiles. Furthermore, it is worth noting that the behavior of some devices might change over time due to aging, temperature and environmental effects. For example, with aging, the battery of a smart phone might show performance degradation in terms of needed energy while charging or it might take more time to charge or might discharge at a faster rate. Similarly, it could happen with other devices like an air conditioner whose usage changes from season to season, also depending on the external temperature. Such behaviors cannot be covered in a limited time trace dataset, which is why there could be some points where the classification fails. As the devices’ behavior or their use can change over the time, a possible future work regards the extension of our model with a patching-based approach for classifiers [40], which focuses on the adaption of existing classification models to new data. As classification often faces scenarios where an already existing model needs to be adapted to a changing environment. Such research work identifies the regions of instance space in which adaptation is needed and, after that, local classifiers for these regions are trained. Such regions after training are incorporated into the model and can handle the predictions even after some changes due to aging or environment. Thus, whenever a decay is detected in the performance of the model, the adaptation is triggered with the goal of finding patches to the classifier that can act efficiently without training the model again from scratch.

## Figures and Tables

**Figure 1 sensors-19-02611-f001:**
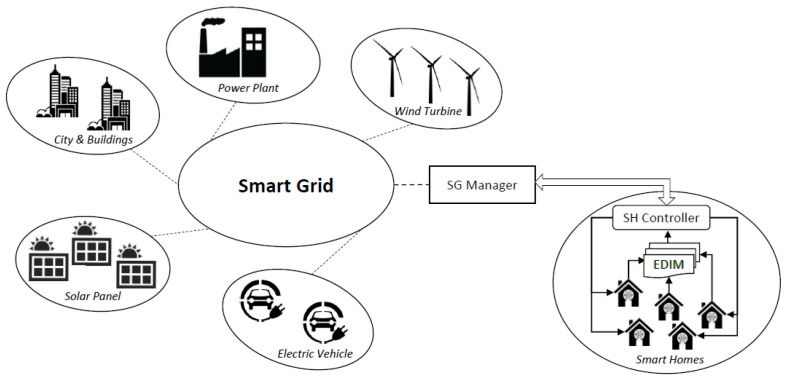
Smart grid overview.

**Figure 2 sensors-19-02611-f002:**
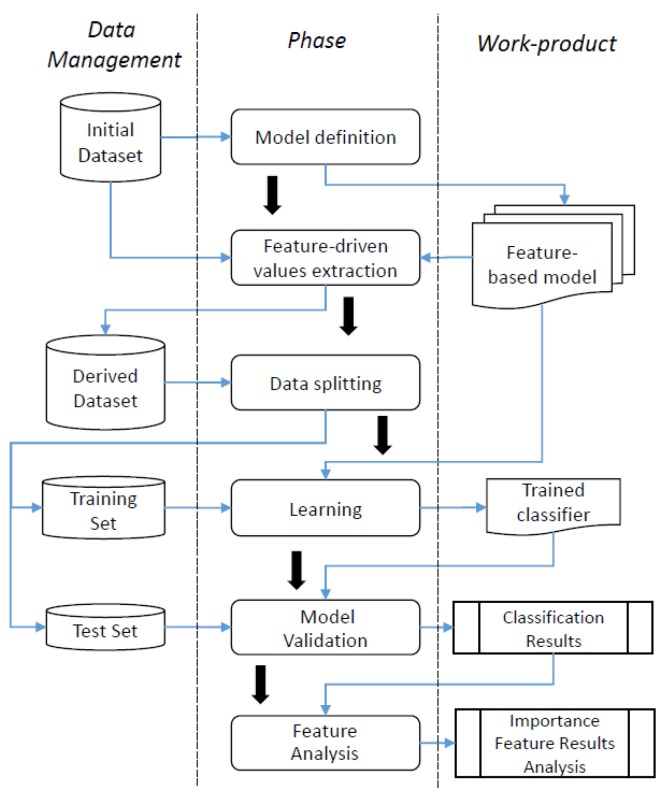
Research approach: data management, phases and work-products.

**Figure 3 sensors-19-02611-f003:**
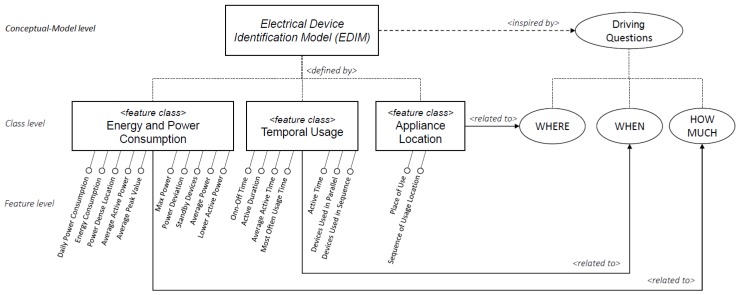
The Electrical Device Identification Model and related questions.

**Figure 4 sensors-19-02611-f004:**
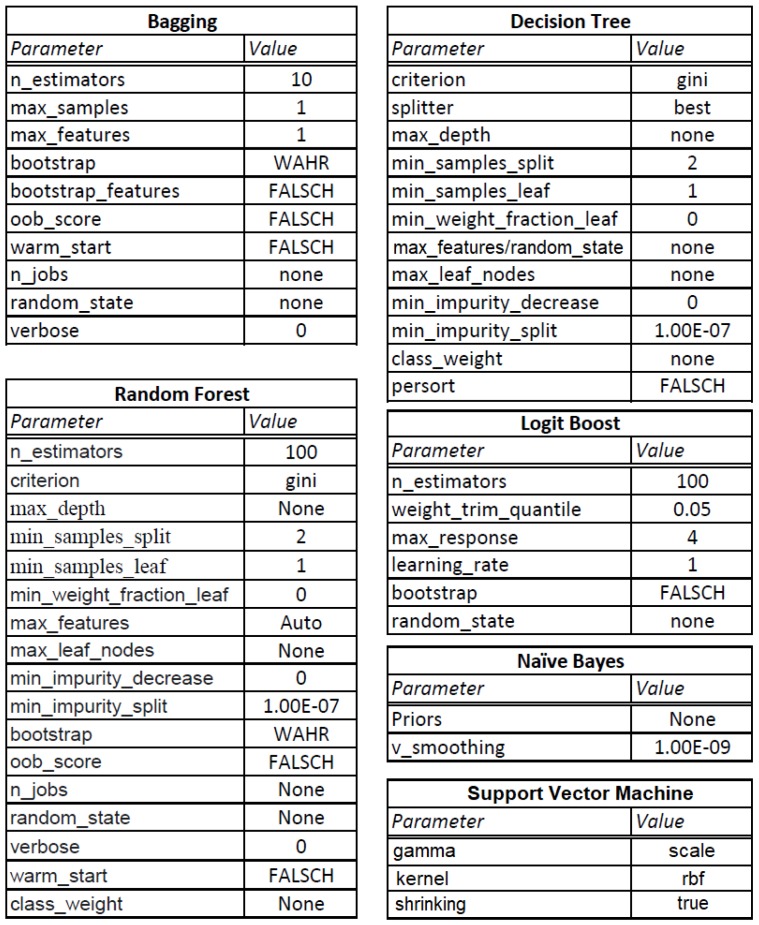
Machine learning algorithms and related parameters.

**Figure 5 sensors-19-02611-f005:**
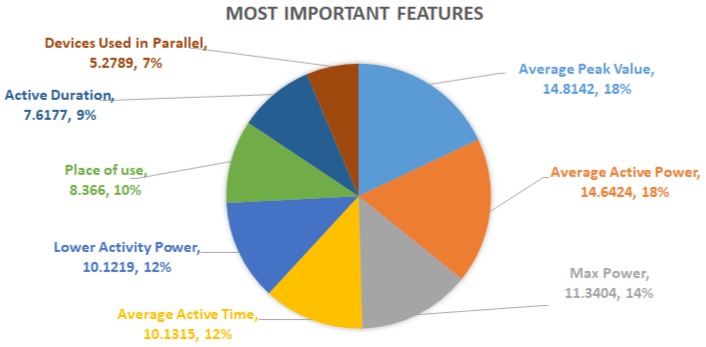
The most important features along with their absolute and relative importance values.

**Table 1 sensors-19-02611-t001:** Comparison of the related works.

Related	Main	Additional	Accuracy	Adopted	Trace
Work	Parameters	Devices	(%)	Approach	Type
[13]	Active and Reactive Power	None	94–97	Not	Not
	Phase shift, $V_{rms}$, $I_{rms}$			specified	specified
[15]	Power consumption,	Measurement and	95.5	Distributed	DT
	Working schedule	Actuation Units			
[16]	Power consumption	Plug-based	85	Distributed	DT
	at low frequency	low-end sensor			
[19]	Power consumption	NIALM device	Not reported	Centralized	AT
[18]	Power factor	Smart Plug	Not reported	Distributed	DT
	Harmonic distortion				
[20]	Active power, Reactive power,	Zigbee Monitor	95	Centralized	AT
	Phase shift, Signature length,				
	Root mean square voltage,				
	Sampling frequency				
[21]	Electrical Noise	Oscilloscope, Laptop	85–90	Centralized	AT
		Custom Data Collector			
[22]	Active power, Reactive power,	Smart Plug	93.6	Distributed	DT
	Root mean square voltage,				
	Phase shift				

**Table 2 sensors-19-02611-t002:** Accuracy of the different classifiers.

#	Algorithm	Accuracy [%]
1	Random Forest	96.51
2	LogitBoost	94.99
3	Bagging	93.02
4	Decision Tree	91.10
5	Naive Bayes	90.26
6	Support Vector Machine	90.11

**Table 3 sensors-19-02611-t003:** Accuracy classification for each appliance provided by the Random Forest.

Electrical Device	True Positive	False Positive	Precision	Recall
Alarm Clock	1.0	0	1.0	1.0
Amplifier	1.0	0	1.0	1.0
Bean to cup	1.0	0	1.0	1.0
Coffee machine	1.0	0	1.0	1.0
Dishwasher	1.0	0	1.0	1.0
Desktop PC	1.0	0	1.0	1.0
Dryer	1.0	0	1.0	1.0
DVD	0.99	0.001	0.941	0.99
Ethernet	0.95	0	1.0	0.95
Freezer	1.0	0	1.0	1.0
Iron	0.80	0.002	0.65	0.80
Lamp	0.88	0.002	0.85	0.88
Laptop	0.96	0	1.0	0.96
Mediacentre	0.99	0	1.0	0.99
Microwave	1.0	0.001	0.95	1.0
Monitor-CRT	0.92	0.002	0.93	0.92
Monitor-TFT	1.0	0	1.0	1.0
PlayStation	0.87	0	1.0	0.87
Printer	1.0	0.001	0.98	1.0
Projector	0.97	0	1.0	0.97
Refrigrator	1.0	0	1.0	1.0
Router	1.0	0	1.0	1.0
Stove	1.0	0	1.0	1.0
Toaster	1.0	0.001	0.95	1.0
TV-CRT	1.0	0	1.0	1.0
TV-LCD	1.0	0	1.0	1.0
TV-REC	0.96	0	1.0	0.96
USB Harddrive	1.0	0	1.0	1.0
Vacuum cleaner	1.0	0	1.0	1.0
Water Fountain	1.0	0	1.0	1.0
Water Kettle	0.57	0.003	0.58	0.57
Wash Machine	1.0	0.002	0.983	1.0
Xmas Lights	0.99	0	1	0.99
**Weighted average**	0.9651	0.0004	0.964	0.9651

**Table 4 sensors-19-02611-t004:** Descending order of features, for level of importance, expressed as a percentage.

#	Feature Name	Importance (I) [%]
1	Average Peak Power	14.8142
2	Average Active Power	14.6424
3	Max Power	11.3404
4	Average Active Time	10.1315
5	Lowest Active Power	10.1219
6	Place of Use	8.366
7	Active Duration	7.6177
8	Devices Used in Parallel	5.2789
9	Average Power	4.9267
10	Most of Usage Time	4.5993
11	Standby Devices	2.0524
12	Active Time	1.9884
13	Power Deviation	1.7767
14	Devices Used In Sequence	1.2207
15	Power Dense Location	0.6378
16	Sequence of Usage Location	0.2529
17	Daily Power Consumption	0.2319
18	Energy Consumption	0.0002
19	On-Off Time	-
**Tot.**	-	100

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
