# Peer review of "A Feature-Based Model for the Identification of Electrical Devices in Smart Environments"

_sensors, 2019, doi:10.3390/s19112611_

Round 1
Reviewer 1 Report
The paper analyses application of different machine learning techniques to device identification in smart home environment based on its electricity consumption footprint.
The introduction requires some revision since power saving is not really a major reason for identification of electric devices in a smart home environment. I would say that the major reason is actually the whole functionality of a smart home.
The approach used is rather strange. The authors seem to apply "classic" models of different some machine learning techniques and check if they perform well on a given dataset. Here one may have a lot of questions. E.g., why neural networks were not used? For each of the applied techniques there are a number of variations/improvements designed for different types of tasks, why did not you apply any? Which parameters and learning algorithms were used in the experiment? All this questions (mandatory for a research) remain unanswered.
The paper has to be significantly re-written starting from the research design stage.
Generally the English is fine, however some proofreading is required due to some inaccuracies in the text, e.g.:
2-40 "on the basis" is duplicated
2-57 "the electrical socket" -> "an electrical socket"
5-186: "that are might have" -> "that might have"
5-186: "the data that on which" -> the data, on which"
Author Response
Dear Reviewer,
thank for your comments that helped us to improve enormusly the article.
Please find enclosed my letter with the overall observations and the related anwers and integration that we provided in the paper..
I hope you will find them sufficient for the publication of the article in this journal.
Best Regards,
Andrea Tundis

Reviewer 2 Report
The paper proposes a set of features for appliance identification. The authors use public traces of 1-second appliance energy use and apply a library of standard ML algorithms. The results show 96.51% accuracy, compared to the usual 85-95% accuracy. Since the Random Forest algorithm performed best, the discussion was based on this method. Table 3 explores the features providing the most information, which is useful information. The paper is well written with a few minor errors:
The title has inconsistent capitalization
Some informal language such as the abstract: "need to face with"
Multiple citations should be combined [1][2] -> [1,2]
Ref 18 has no author
Figure 3 is not easy to read, consider only plotting 80-100% to show more detail
There have been prior attempts to generalize features for appliance signatures and at one point a public database was created (containing many more features than 1-s energy). These works were published in BuildSys, they should be included and compared.
Table 1 provides high-level accuracy results in several works. However, it's not clear if a fair comparison to the given work is possible since many prior works used vastly different features and test environments. It is good to have this information but the limitations on comparison should be more clear. For example, some of the prior works use aggregate traces of multiple devices and attempts to disaggregate energy usage. This is a harder problem than working with single device traces. Maybe you could remove other works that aggregate devices?
The experimental approach is not clear (Section 5). The public dataset contains single device traces. You list devices in Table 2. Were these the set of devices used in your tests? Is this all of the devices in the dataset or do you remove some? I assume you only consider single devices and do not aggregate multiple devices to test disaggregation. Explain exactly how you construct the test/training sets. A discussion of limitations should be included. For example, the trace data has a TV-LCD but there are a myriad of TV-LCDs. Your method would have to train on specific models and not be able to generalize to these device categories. How much training data is needed to classify a device? In practice, how could you collect this data? Also, the trace data is relatively short. Could the behavior of devices change over time (aging, temperature/environmental effects?)? How could your model compensate for this?
Author Response

(The authors gave the same response as above.)

Round 2
Reviewer 1 Report
The authors have addressed the indicated issues. The paper looks much better now and can be accepted in its present form.
Author Response
I would like to thanks the reviewer about that.
Best Regards,
The Authors
Reviewer 2 Report
The paper is generally improved. However, there are still a few cases where the writing [English] needs to be improved:
34: difficult the energy management
86: 95.5% of accuracy.
431: 80% of accuracy
I am sure with a careful review you can find more examples.
On line 144 the sentence begins with "Basically, the more examples.." should be rewritten. It sounds like you are saying that with more training data the performance will increase. This is not necessarily true. For example, confounding data could reduce the performance of an ML algorithm. Possibly this sentence can be removed or at least rewritten to be more correct.
On line 448 you claim globally better performance compared with other related works. This comparison is not fair. To make this claim you must implement the other algorithms and test them on the same dataset using the same test procedure. Your dataset is fairly limited and some of the related works operate on more complex datasets. So, saying your results are better could be an artifact of the dataset you tested on and not necessarily the approach. This claim is repeated on line 506 as well.
Figure 5 is not needed. You are showing 5 scalar values, use a table sorted by accuracy instead.
Table 2 would be easier to read if the rows were sorted alphabetically by "Electrical Device". You mentioned the water kettle in the text and it took me a while to find it in the table because they are arbitrarily listed.
Author Response
LETTER TO THE EDITOR
May 31st, 2019
Dear Editor,
We would like to thank the reviewers for giving us precious suggestions in the reviewing phases of our paper “Electrical Devices Identification in Smart Environments Driven by Features and Based on Machine Learning”.
We are glad to inform you that all the issues raised by the reviewers have been addressed as well as all of the recommended changes and suggestions have been implemented in the paper respectively. In the following, each of the concerns raised by the reviewers is reported in point-form with our answers and the description of the possibly revisions made to the paper (highlighted in green color in the paper). Moreover, other minor but significant changes have been made throughout the rest of the paper to improve its readability.
We hope you will find the paper improved in its content and presentation and hope you will deem it suitable for publication in the Special Issue "Smart Monitoring and Control in the Future Internet of Things" of the journal Sensors.
Yours sincerely,
The Authors
reviewer 1
No further comments
reviewer 2
The paper is generally improved. However, there are still a few cases where the writing [English] needs to be improved:
Observation 1 - 34: difficult the energy management.
ü Answer to Observation 1 - We would like to thanks the reviewer for such suggestion, we addressed it by re-writing the sentence and clarifying it at Page 1 – Page 2 : rows 33-35. The text is reported below.
“Typically, the traditional homes have devices that work locally and manually, usually by switching them on/off by pushing a button, with a limited control in terms of their automatic management.”
Observation 2 - 86: 95.5% of accuracy, 431: 80% of accuracy
Answer to Observation 2 - We would like to thanks the reviewer for having noticed such mistake. We have corrected them as following:
- 85: 95.5% of accuracy. à95.5% accuracy.
- 431: 80% of accuracy à 80% accuracy
Observation 3 - I am sure with a careful review you can find more examples.
ü Answer to Observation 3 - We would like to thanks the reviewer the suggestion and we identified and corrected other similar expressions, such as the one below reported:
- 426: 96.51 % of accuracy à 96.51% accuracy
Observation 4 - On line 144 the sentence begins with "Basically, the more examples.." should be rewritten. It sounds like you are saying that with more training data the performance will increase. This is not necessarily
true. For example, confounding data could reduce the performance of an ML algorithm. Possibly this sentence can be removed or at least rewritten to be more correct.
ü Answer to Observation 4 - We would like to thanks the reviewer the comment. We agreed with such suggestion by removing the sentence, from Page 4 rows 143-144, without loss of clarity.
Observation 5 - On line 448 you claim globally better performance compared with other related works. This comparison is not fair. To make this claim you must implement the other algorithms and test them on the same dataset using the same test procedure. Your dataset is fairly limited and some of the related works operate on more complex datasets. So, saying your results are better could be an artifact of the dataset you tested on and not necessarily the approach. This claim is repeated on line 506 as well.
ü Answer to the Observation 5 - thank you for the comment. We added the result of the SVM in order to cover the most popular algorithm and in particular by also highlighting in the article that our result are particularly compared with those presented in the related work [15], as we used the same dataset and a similar feature based approach. For the sake of completeness we reported the updated sentences below.
Page 14- rows 448-451. In summary, we globally obtained very high performances in terms of accuracy compared with other related works, and in particular respect to the related work [15] by only using 19 features instead of 517 features on the same dataset, by requiring both less computational resources and computing time.”
Page 19 – row 504-506. Further experiments and details related to the Random Forest classifier, that provided the highest accuracy equal to 96.51%, have been conducted and discussed, as it outperformed respect to the related work 15 by using only 19 features on the same dataset.
Observation 6 - Figure 5 is not needed. You are showing 5 scalar values, use a table
sorted by accuracy instead.
ü Answer to Observation 6 - Thank you so much for your suggestion. We addressed it by replacing Figure 5 with a table sorted by accuracy (now Table 2) at Page 13.
Observation 7 - Table 2 would be easier to read if the rows were sorted alphabetically by "Electrical Device". You mentioned the water kettle in the text and it took me a while to find it in the table because they are arbitrarily listed.
ü Answer to Observation 7 - Thank you so much for your suggestion. We addressed it by sorting alphabetically the Electrical Devices in the Table (currently Table 3), at Page 14.
